# Study on Potato Bud Cultivation Techniques in a Greenhouse in Spring

**DOI:** 10.3390/plants12203545

**Published:** 2023-10-12

**Authors:** Chaonan Wang, Rui Bao, Hui Zhang, Leijuan Shang, Huilin Wang, Zhongmin Yang, Chong Du

**Affiliations:** College of Horticulture, Xinjiang Agricultural University, Urumqi 830052, China; wcn0107@126.com (C.W.); 18195951010@163.com (R.B.); 18160575357@163.com (H.Z.); 18709945585@163.com (L.S.); wanghuilin@126.com (H.W.); yangzhongmin161220@126.com (Z.Y.)

**Keywords:** potato, sprout planting, cultivation techniques, growth, yield and quality

## Abstract

The species degeneration caused by traditional potato cultivation methods is becoming increasingly evident, and it is particularly important to study new potato cultivation methods. Sprout planting technology has the advantages of large reproductive capacity, fast growth speed, and simplified maintenance of cultivated crops. In this study, four disease-free potato varieties (‘Fujin’, ‘Youjin’, ‘Zhongshu 4’, and ‘Feiwuruita’) were treated with different parts (top bud, middle bud, and tail bud) and different bud lengths (10 cm, 15 cm, and 20 cm), and then potato sprout planting was carried out. A nutrient pot experiment was performed following a randomized complete block design (RCBD) with various replicates and a natural control (CK) treatment. By comprehensively measuring the emergence, chlorophyll content, net photosynthetic rate, dry matter distribution during the bulking period of blocks, and effect of growth and quality with bud direct seeding under both treatments, it was found that potato block top bud direct seeding cultivation is significantly superior to other parts. In terms of early maturity and yield statistics, the advantage of top bud cultivation in ‘zhongshu 4’ is most obvious; it reaches maturity an average of 14 days earlier, and the yield can be increased by 38.05%. Therefore, top bud direct seeding is more suitable for potato sprout planting technology. On this basis, the 20 cm and 15 cm bud length treatments of top buds were used for direct cultivation, and all the above indicators performed well. Among them, in the zhongshu 4 variety, the yields of 15 cm and 20 cm bud length treatments increased by 41.78% and 38.05%, the growth rates of commercial potatoes increased by 6% and 6.9%, respectively, and the effects were the most obvious. In conclusion, the deep research and application of potato sprouting technology has high utilization value for improving potato yield and quality and has guiding significance for greenhouse potato cultivation in early spring.

## 1. Introduction

Potato (*Solanum tuberosum* L.) is a well-known tuber crop consumed by approximately 1.3 billion people as a staple food [1]. And it is also used as an important industrial raw material for the production of starch, ethanol, and animal feed [2]. At present, the protected cultivation of potato mainly involves the direct seeding of tubers, which has the crucial disadvantages of high seed consumption and a low reproduction coefficient. Planting potato sprouts is a method which uses the young buds from potato tubers as “seeds” for sowing and planting (Appendix A). This approach increases reproductive output, reduces reproduction time, ensures plant diversity in crop cultivation, and is applied to both traditional Chinese medicine purposes and vegetable crops [3,4,5,6].

As early as the 1960s, China conducted preliminary research on the effects of potato sprout planting. The researchers placed the seed potatoes in room temperature (25 °C) and dark conditions to cultivate buds. After the buds were approximately 15 cm long, they were broken at the base and planted in loose and high-moisture soil. The growth node of the buds was buried 1–2 cm from the ground surface [7]. Potatoes were sprouted in a semi-humid soil and germinated at a suitable temperature of 20–25 °C, reaching a germination length of 3.5–10.5 cm after 8 days, and had a relatively high transplantation survival rate [8]. Sprouting potatoes can also be transplanted separately. When the seedling height is 6–10 cm, 5–7 leaves can be cut from the base, and the seedling can be rooted and planted. The yield is higher than that of traditional potato block direct seeding, and the commercial potato has good properties [9]. In fact, the older the bud age of potato seeds, the earlier the yield and the greater the early yield [10]. Sprouting potatoes have a longer underground stem and more underground stem nodes, which greatly increases the tuber-setting surface, thus increasing the number of tubers. Research has found that in autumn potato cultivation, direct seeding with a sprouting length of 2.5 cm can increase the highest yield of potatoes by 89.4% compared to nonsprouting methods [11]. 

Potato sprout planting can also save the amount of seeds potatoes need. Traditional potato block direct seeding uses 1500–2250 kg/ha of potato seeds, but sprout planting requires only 300–375 kg/ha, which can save nearly five times the amount of seed potatoes [12]. In addition, using potato sprout planting technology for cultivation can not only restore seed quality but can also achieve a good detoxification effect and strong disease resistance in the early stage, which can greatly reduce the occurrence of diseases and pests. For example, planting sprouts can help reduce potato late blight and reduce the probability of infected plants [13]. Using seed potatoes when the bud height is approximately 5 cm and accompanied by many fibrous roots and breaking them off from the seed potatoes for transplantation to grow seedlings can increase the reproductive coefficient of virus-free seed potatoes [14,15]. Through research on potato seedling cultivation and transplantation, it was found that the plant morphological and physiological growth indices were significantly different among the treatments, and the 7 cm potato bud length treatment was proven to be the best treatment for raising seedlings under simulated field conditions [16].

At present, the protective cultivation of potatoes mainly involves direct sowing of tubers, which has the key drawbacks of high seed consumption and a low reproductive coefficient. But using potato sprouts for direct cultivation can effectively reduce seed consumption, improve reproductive efficiency, and improve the overall efficiency of crops. However, there are few reports on direct cultivation of potatoes through potato sprouts. This experiment demonstrates the adaptability and feasibility of directly cultivating potatoes with different lengths of potato buds. The results showed that potato sprout cultivation is beneficial for reducing the amount of seeds potatoes used, increasing the reproductive coefficient, shortening the growth period, increasing yield, and achieving the goal of early maturity and high yield. This study aims to explore the feasibility and adaptability of direct facility cultivation of potato sprouts, providing valuable guidance for potato production.

## 2. Results

### 2.1. The Effect of Bud Planting on the Emergence Rate of Potatoes

According to Table 1, which shows the experimental results of direct planting potato buds from different locations, the emergence rate of buds is as follows: direct planting of potato top buds > middle buds > potato bud blocks (CK) > tail buds. Treatments varied significantly among the four varieties. The emergence rate of ‘Fujin’ and ‘Youjin’ potato top buds after treatment showed the greatest improvement compared to their respective controls, with increases of 3.95% and 3.72%.

The emergence rate of potato top buds of varying lengths selected for planting experiments is shown in Table 2. The results showed that the direct planting treatments with a bud length of 15 cm > 10 cm > 20 cm > direct seeding with a bud block (CK), showing significant differences within the same variety, and the performance of the four tested varieties was consistent. In the direct seeding with a 15 cm shoot length, the emergence rate of ‘Feiwuruita’ was the highest, reaching 98.56%. The emergence rate of ‘Fujin’ was 98.18%, although not as high as that of ‘Feiwuruita’. ‘Fujin’ had the largest increase compared to the control, reaching 3.95%.

### 2.2. The Effect of Bud Planting on the Growth of Potatoes

The direct planting of different parts of potato buds revealed differences in these indicators during the seedling stage among treatments of the same variety (Table 3). The measured values from high to low were as follows: direct planting of potato top buds > middle buds > tail buds > potato bud blocks (CK), and the performance of all tested varieties was consistent. Within the same variety, there were significant differences in the above indicators among the three treatments compared to the control. The highest performance in plant height and number of compound leaves were 12.08 cm and 7.38 leaves observed in the ‘Youjin’ potato with a top bud direct planting treatment. The maximum stem diameter was 10.84 mm in the potato top bud direct planting treatment of ‘Zhongshu 4’.

During the period of potato block enlargement, the growth indicators of plant height and stem diameter in different parts of the potato bud direct planting treatment were as follows: potato top bud direct planting > middle bud > tail bud > potato bud blocks (CK), while in the number of compound leaves, potato bud blocks (CK) > tail bud, and the tested varieties showed consistent performance. Among them, the highest plant height and highest number of leaves were observed in the top bud treatment of the ‘Youjin’ potato, which were 78.12 cm and 32.56, respectively. The largest stem diameter was 16.80 mm, which was observed in the top bud treatment of ‘Zhongshu 4’ potatoes.

Top potato buds of different lengths were used for direct planting (Table 4). During the period of potato block enlargement, each treatment significantly differed among the growth indicators in terms of direct planting with a top bud length of 20 cm > 15 cm > 10 cm > direct seeding with a bud block (CK). There were significant differences in plant height and leaf number among the different treatments, and there was no significant difference in stem diameter measurements between the 15 cm and 20 cm treatments. The plants with the best performance in terms of plant height and stem diameter were the 20 cm direct planting treatment of the ‘Feiwuruita’ variety, which were 81.56 cm and 17.43 cm, respectively; the ‘Youjin’ potato with the largest number of leaves had 33.27 leaves under the same treatment.

During the seedling stage, growth indicators such as plant height, stem diameter, and number of compound leaves showed a trend from high to low: direct planting with a bud length of 20 cm > 15 cm > 10 cm > direct seeding with a bud block (CK). There were significant differences within the varieties, and the performance of each variety was consistent. Among them, for the under-20 cm bud planting, the highest plant height was 13.14 cm in ‘Youjin’, the maximum stem diameter was 12.06 mm in ‘Zhongshu 4’, and the largest number of compound leaves was ‘Fujin’, which was 7.21 pieces.

### 2.3. Effects of Bud Planting on Chlorophyll Content and Net Photosynthetic Rate of Potatoes

During the seedling stage, both indicators of the leaves of the four tested varieties showed a decreasing trend from top bud direct planting > middle bud > tail bud > bud block direct seeding (CK), with significant differences among the treatments (Table 5). Among them, the highest chlorophyll content was the top bud direct planting of ‘Zhongshu 4’, which was 2.84 mg/g, and the minimum value for the ‘Youjin’ potato bud block seedlings (CK) was 1.32 mg/g. The highest photosynthetic rate of the ‘Feiwuruita’ variety under direct cultivation of potato top buds was 14.62 μmolCO_2_·m^−2^·s^−1^, and the lowest was 10.14 μmol CO_2_·m^−2^·s^−1^ in the control treatment for the ‘Youjin’ variety. All varieties showed the same performance.

During the period of potato block enlargement, both indicators of the four tested varieties under direct cultivation of different parts of buds sequentially increased from high to low, followed by direct cultivation of potato top buds > middle buds > tail buds > direct seeding of potato bud blocks (CK). The highest chlorophyll content was 4.52 mg/g in the top bud direct planting treatment of ‘Zhongshu 4’, and the highest photosynthetic rate was 19.7 μMolCO_2_·m^−2^·s^−1^ in the top bud direct planting treatment of ‘Feiwuruita’. The minimum values all appeared in the control treatment of ‘Youjin’ potatoes, which were 2.80 mg/g and 15.80 μMolCO_2_·m^−2^·s^−1^, respectively.

Top potato buds of different lengths were used for direct planting (Table 6). During the seedling stage, the chlorophyll content and photosynthetic rate of potato leaves showed a trend of 20 cm direct planting > 15 cm > 10 cm > bud block direct seeding (CK). There were significant differences among different treatments within the varieties, and the performance of the four varieties was consistent. The highest chlorophyll content was 3.07 mg/g in the 20 cm seedling treatment of the ‘Zhongshu 4’ variety, and the lowest was 1.32 mg/g in the bud block direct seeding (CK) of the ‘Youjin’ variety. The highest photosynthetic efficiency was observed in the ‘Feiwuruita’ variety with a 20 cm direct planting treatment of 15.84 μMolCO_2_·m^−2^·s^−1^, with the lowest being the control treatment of the ‘Youjin’ variety of 10.14 μMolCO_2_·m^−2^·s^−1^.

During the period of potato block enlargement, both indicators were measured from high to low as 20 cm direct planting > 15 cm > 10 cm > bud block direct seeding (CK) treatments. The three direct planting treatments showed significant differences from the control; the performance trends of the four varieties were the same. The highest chlorophyll content was 4.74 mg/g in ‘Zhongshu 4’, and the lowest was 2.80 mg/g in ‘Youjin’ compared with the control treatment. The highest photosynthetic rate was 21.70 μmolCO_2_·m^−2^·s^−1^ in ‘Feiwuruita’ with direct planting of 20 cm buds, and the minimum was 15.82 μmolCO_2_·m^−2^·s^−1^ in ‘Youjin’ compared to the control treatment. The tested varieties were consistent.

### 2.4. Distribution of Dry Matter during the Bulking Period of Potato Block in Bud Planting

The dry matter allocation rate of potato plants directly cultivated from potato buds taken from different parts during the potato block expansion period is shown in Figure 1. The dry matter allocation rate of tubers was ranked from high to low in the following order: potato top bud direct planting > middle bud > tail bud > bud block direct seeding (CK). The tested varieties showed the same performance, with ‘Zhongshu 4’ having the highest allocation rate of 38.9% for tubers and ‘Feiwuruita’ having the lowest rate of 20.9%. The dry matter allocation rate of stems and leaves was highest in the bud block direct seeding treatment, followed by the potato tail bud > middle bud > top bud direct planting. The overall allocation rate of stems and leaves in the control treatment of each variety was above 74.5%.

The distribution rate of dry matter in potato plants during the expansion period of potato block enlargement with different lengths of potato top buds directly planted is shown in Figure 2. The distribution rate of dry matter in potato tubers was as follows: 20 cm direct planting treatment > 15 cm > 10 cm > bud block direct seeding (CK). The performance of all tested varieties was consistent, with potato tuber distribution rates being the highest in the 20 cm and 15 cm direct planting treatments, with percentages ranging from 37.8 to 40.7%, differing by only approximately 3%. The distribution rate of dry matter in stems and leaves is ordered as follows: bud block direct seeding (CK) > 10 cm > 15 cm > 20 cm direct planting, and their control treatments were all above 70%. The maximum tuber allocation rate of ‘Zhongshu 4’ directly planted at 20 cm was 40.7%, while the lowest was only 24.1% in the control treatment of ‘Youjin’.

### 2.5. The Effect of Direct Planting of Potato Buds on Early Maturity and Yield of Potatoes

The effects of cultivating with potato buds from different parts on the early maturity and yield of potatoes are shown in Table 7. Each treatment was planted at the same time, and the harvest time and growing days within the same variety were different. The growing days gradually increased according to top bud > middle bud > bud block direct seeding (CK) > tail bud. The growing days of the top bud direct-cultivation treatment were the shortest, with the earliest maturation being the top bud treatment of ‘Zhongshu 4’, which lasted for 55 days. The latest harvest was the ‘Youjin’ tail-bud treatment, which took 78 days. The top-bud cultivation of ‘Fujin’ had the most early relative growing days, which was 15 days earlier than that of its control.

The correlation analysis of potato yield indicators for different parts of the potato bud direct planting treatment is shown in Table 8. In terms of average single potato weight, there were significant differences between the top bud direct planting treatment and the middle bud direct planting treatment compared to the control. In terms of average plot yield and equivalent yield of the same variety, within the same variety, the order was potato top bud direct planting treatment > middle bud > tail bud > bud block direct seeding (CK). The highest yield was 4.42 kg/ha × 10^4^ kg/ha in the ‘Zhongshu 4’ with potato top bud direct planting treatment, and the lowest was 3.03 kg/ha × 10^4^ kg/ha in the ‘Feiwuruita’ with bud block direct seeding treatment; the most significant increase in relative yield was observed in the ‘Zhongshu 4’ with potato top bud direct planting, which increased yield by 38.05%.

The effect of different lengths of directly seeded potato top buds on the early maturity and yield of potatoes (Table 9) is as follows: All tested varieties were planted on the same day, and the harvest time and growing days varied among different varieties. The growing days between treatments within the same variety were characterized by the longest duration of bud block direct seeding > 10 cm > 15 cm > 20 cm direct planting. The maximum number of days for early maturing of 20 cm potato buds from ‘Fujin’ and ‘Youjin’ was 16 days.

Compared with the control, there were differences in average single potato weight, average plot yield, and equivalent yield in the treatment of direct planting with different lengths of potato top buds (Table 10). Among them, the yield was highest according to 20 cm direct planting > 15 cm > 10 cm > bud block direct seeding (CK). The experimental varieties performed the same. The highest yield was obtained from the 20 cm direct planting treatment of ‘Zhongshu 4’, with a yield of 4.54 × 10^4^ kg/ha. The lowest yield was the ‘Feiwuruita’ variety, which uses potato bud block seedlings at 3.03 × 10^4^ kg/ha. The potato with the highest relative yield increase was ‘Zhongshu 4’, with an increase of 41.78%.

### 2.6. The Effect of Different Potato Bud Direct Planting Treatments on the Quality of Potatoes

The results of quality indicators of potato fresh matter showed that there was no significant difference in starch, reducing sugar, crude protein, or Vc content among different varieties, regardless of whether they were directly seeded with different lengths of buds or different parts of tubers (Table 11 and Table 12). However, there was no significant difference in starch content between the ‘Zhongshu 4’ and ‘Feiwuruita’ varieties under the two treatments, and the same condition was also found in the Vc content of ‘Fujin’ and ‘Yujin’. The reducing sugar and crude protein contents showed significant differences among varieties. In addition, the yield of commercial potatoes showed a trend of 15 cm direct planting > 20 cm > 10 cm > bud block direct planting (CK), with top bud direct planting > middle bud > tail bud > bud block direct planting (CK). The two treatments of ‘Zhongshu 4’ both showed the highest increase in yield of commercial potatoes, with an increase of 6.9%.

## 3. Discussion

In this experiment, a potato sprout planting method was used to test four potato varieties, and it was found that different bud locations and bud length treatments could significantly improve potato growth, quality, and yield-related indexes. This method greatly improves the reproductive coefficient of potatoes, which is consistent with previous research results on improving the utilization efficiency of potato seeds through repeated seedling breakage and transplantation [17,18].

Using different parts of the potato buds for direct planting, the emergence rate reached over 90%, among which the top bud emergence rate was as high as 98%, while the tail bud emergence rate was significantly lower than that of the bud block direct seeding treatment, showing a clear advantage of apical dominance. From the seedling stage to vigorous growth, the growth trend of the middle and tail bud treatments was delayed longer than that of the top bud, while the top bud direct cultivation treatment showed advantages such as vigorous growth, more leaves, strong photosynthetic ability, and higher yield, which is consistent with the principle that potato tubers have the strongest physiological and genetic potential at the top [19]. During the process of planting potato sprouts, ensuring the integrity of potato sprouts and their basal root primordia for planting is beneficial for the seedling growth rate, which is related to the occurrence and quantity of potato sprout root primordia [20]. Top bud direct planting can also improve the emergence rate [21,22], increase yield [23] by facilitating early leaf expansion, and accelerate growth of potato top-buds [24].

This study selected longer potato sprouts (20 cm and 15 cm) for direct seeding, which can achieve higher leaf numbers, photosynthetic efficiency, and potato yield. This further supports that potatoes with many root primordia, well-developed root systems, and more stolons can produce more tubers in a given soil profile and increase yield [25]. Cultivating with the 20 cm long potato sprouts resulted in more leaves and greater weight, which is consistent with the changes in endogenous hormone content in potato sprouts at different physiological ages [26]. 

Yield in the direct planting of potato buds of different lengths and different parts, except for the tail-buds, increased. The reason is that the underground stem node is longer, the root system is developed, the growth rate is fast, the plant leaf area is large, and the growth is vigorous [27], which promotes the plant to enter the potato block expansion period earlier, so much so that in the early stage of potato block growth and development under relatively low temperatures, the aboveground dry matter of the plant is transported underground, thereby promoting the growth of the tuber [28]. This is consistent with the research outcomes of potatoes that are tolerant to low temperatures and where stem and leaf assimilates are transported to the tuber during potato block development [29]. Potatoes have a certain prematurity, and in this study, potato buds cultivated with top-buds and 15 cm and 20 cm potato buds planted directly showed early maturity, which may be related to the fast growth rate of plants in early stages, as well as the large leaf area, strong photosynthetic rate, and the early entry of the potato block expansion stage [30]. However, the growth period of potato tail-buds was prolonged, which was basically in line with the apical dominance principle, but the specific reason should be further investigated. 

## 4. Materials and Methods

### 4.1. Plant Materials and Growth Conditions

The tested potato varieties are ‘Fujin’, ‘Youjin’, ‘Zhongshu 4’, and ‘Feiwuruita’, all of which are virus-free seed potatoes, and the growth period of the four varieties is the same; all belong to the precocious varieties, which were collected from the Jilin Academy of Agricultural Sciences, China. These materials yield healthy seed potatoes with excellent phenotypes, such as plant height (50~72 cm), average number of potatoes per plant (5~7), average potato weight (115~151 g), and commercial potato rate (71.3%~88%). The experiments were conducted from March to July in 2022 and 2023, respectively, and planting was carried out in a greenhouse. The tested soil is humus soil, with the following physical and chemical properties in the 0–30 cm profile: total nitrogen 1.41 g/kg [31], total phosphorus 0.34 g/kg [32], total potassium 8.9 g/kg [33], and organic matter 40.6 g/kg [34].

### 4.2. Potato Bud Treatment

(1)Direct cultivation of potato buds with varying lengths: There were three potato bud length treatments: 10 cm, 15 cm, and 20 cm. Virus-free seed potatoes that were of the same size, robust, and beyond the dormancy period were selected, and germination was induced under dark conditions. When the length of the potato buds reached 10 cm (16–18 d), 15 cm (22–24 d), and 20 cm (28–30 d) (Appendix A), the top buds were broken off from the base of the potato and kept moist.(2)Direct cultivation of potato buds broken off from different positions: Three treatments were established. The positions included the top bud (the bud at the top of the seed potato), middle bud (the bud in the middle of the seed potato), and tail bud (the bud at the bottom of the seed potato, also known as the base bud) of the potato (Appendix A). When the length of the potato bud was 15 cm (22–24 d), the bud was broken from the base of the potato, and the broken potato bud received the same treatment as in (1).(3)Direct sowing of buds: The treatment of seed potatoes was the same as in (1). When the buds were 1–2 cm long, the buds were grown under scattered light for 2–3 days, and the seed potatoes were cut into bud tubers weighing approximately 20 g [35].

After the potato buds in treatments (1) and (2) have been removed for a long time, they will lose a substantial amount of water and cannot survive. Therefore, the broken potato buds should be immediately moistened, planted, and watered in a timely manner to ensure that the potato buds do not dehydrate. The three experimental treatments adopted high-bed cultivation, with a bed width of 1.2 m, a bed length of 5 m, and a bed height of 10 cm, covered with transparent plastic film. Double-row cultivation was performed in a random block arrangement.

### 4.3. Measurement of Various Indicators for Potato Sprouting

After the end of the seedling stage, 5 plants were taken from each treatment for measurement of growth indicators. After transplanting, 3 plants were then randomly selected from each plot every 7 days for measurement of each indicator. Quality indicators were uniformly measured after harvest. The specific measurement items and methods are as follows:(1)Determination of growth indicators included [root-to-shoot ratio] = underground dry weight/aboveground dry weight; [strong seedling index] = (stem diameter/plant height + underground dry weight/aboveground dry weight) × whole plant dry weight; chlorophyll content was determined by the mixed extraction method using acetone and ethanol; root activity was measured using the TTC (triphenyltetrazolium chloride, TTC) method [31]; and photosynthetic parameters, including net photosynthetic rate, transpiration rate, stomatal conductance, and intercellular CO_2_ concentration, were measured using a portable photosynthetic instrument Li6400. The measured position was the third leaf below the growth node [36]. The above indicator values were averaged over 5 samples.(2)Potato quality assessment through the determination of potato fresh matter quality indicators: these indicators included the determination of crude protein content by Coomassie brilliant blue staining [37]; reducing sugar content was determined by 3,5-dinitrosalicylic acid colorimetry [38]; starch content was determined using the anthrone sulfuric acid method [39]; and the content of vitamin C was determined using the molybdenum blue colorimetric method [40]. And the content of chlorophyll was determined by acetone extraction [41].

### 4.4. Experimental Design and Statistical Analysis

The recorded data were statistically analyzed by using SPSS 20.0 statistical software (SPSS Inc., Chicago, IL, USA). One-way analysis of variance (ANOVA) with post hoc tests (Tukey’s test) was used to detect differences among treatments. The mean values showing significant differences were compared with the Tukey test at a 5% level [42].

## 5. Conclusions

In conclusion, this study found that potato sprouting technology was used for potato facility planting, and the selection of potato buds with 20 cm and 15 cm lengths and at the top of potato blocks had a significant improvement on important indicators such as potato yield, maturity, and quality, which has important theoretical guiding significance for optimizing potato cultivation technology and improving production levels.

## Figures and Tables

**Figure 1 plants-12-03545-f001:**
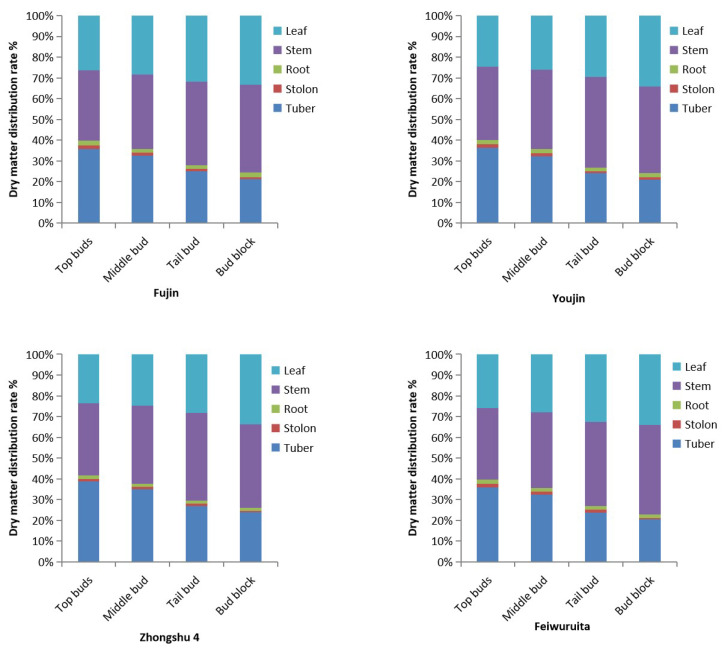
Dry matter distribution rate of different parts of potato block planting during potato block expansion period.

**Figure 2 plants-12-03545-f002:**
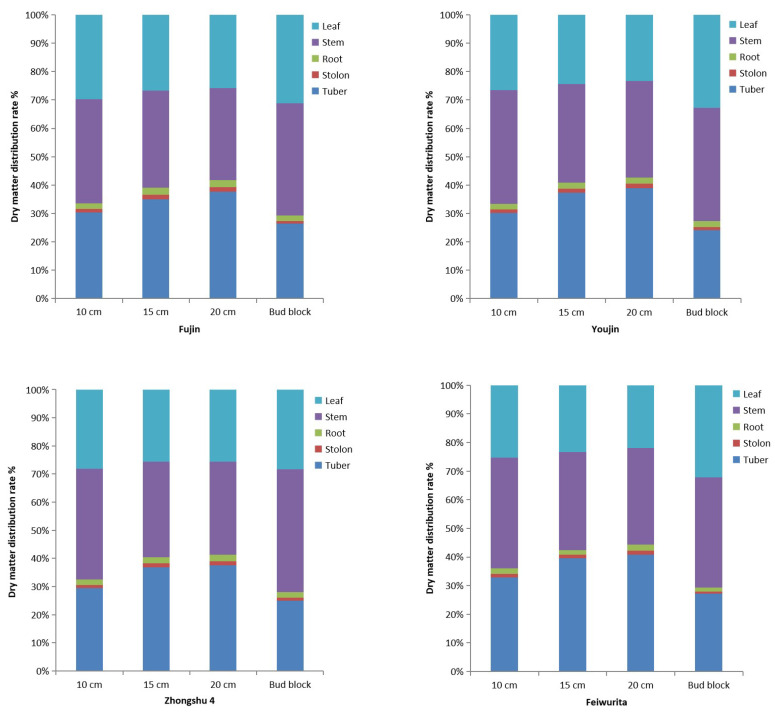
Dry matter allocation rate of different length potato buds planting during potato block expansion period.

**Table 1 plants-12-03545-t001:** The emergence rate of potato buds taken from different locations.

Treatment	Fujin	Youjin	Zhongshu 4	Feiwurita
Top bud	98.18 ± 0.23 ^a^	98.09 ± 0.16 ^a^	98.26 ± 0.14 ^a^	98.56 ± 0.06 ^a^
Middle bud	97.46 ± 0.13 ^b^	97.53 ± 0.20 ^b^	97.40 ± 0.09 ^b^	97.92 ± 0.10 ^b^
Tail bud	92.03 ± 0.17 ^d^	91.07 ± 0.14 ^d^	94.39 ± 0.13 ^d^	93.45 ± 0.10 ^d^
Bud block	94.23 ± 0.17 ^c^	94.37 ± 0.14 ^c^	96.19 ± 0.13 ^c^	96.05 ± 0.10 ^c^

Means followed by a different letter within the column are significantly different at (*p* < 0.05) probability level according to the analysis of variance (ANOVA).

**Table 2 plants-12-03545-t002:** The emergence rate of different lengths of potato buds.

Treatment	Fujin	Youjin	Zhongshu 4	Feiwurita
Bud length 10 cm	97.45 ± 0.16 ^b^	96.44 ± 0.14 ^b^	97.19 ± 0.10 ^b^	97.29 ± 0.08 ^b^
Bud length 15 cm	98.18 ± 0.18 ^a^	98.09 ± 0.13 ^a^	98.26 ± 0.11 ^a^	98.56 ± 0.14 ^a^
Bud length 20 cm	96.27 ± 0.07 ^c^	95.09 ± 0.10 ^c^	96.92 ± 0.14 ^b^	96.84 ± 0.06 ^c^
Bud block	94.23 ± 0.03 ^d^	94.37 ± 0.20 ^d^	96.19 ± 0.20 ^c^	96.05 ± 0.06 ^d^

Means followed by a different letter within the column are significantly different at (*p* < 0.05) probability level according to the analysis of variance (ANOVA).

**Table 3 plants-12-03545-t003:** Effects of different potato bud parts on plant growth.

Treatment		Plant Height (cm)		Stem Diameter (mm)		Number of Compound Leaves (Piece)	
		Seedling Stage	Expansion Period	Seedling Stage	Expansion Period	Seedling Stage	Expansion Period
Fujin	Top bud	10.27 ± 0.10 ^e^	67.89 ± 0.19 ^i^	10.14 ± 0.11 ^cd^	16.74 ± 0.11 ^a^	6.34 ± 0.14 ^d^	29.36 ± 0.19 ^cd^
	Middle bud	10.02 ± 0.10 ^f^	65.90 ± 0.13 ^j^	9.88 ± 0.09 ^de^	16.07 ± 0.11 ^c^	6.17 ± 0.09 ^de^	28.74 ± 0.20 ^efg^
	Tail bud	9.25 ± 0.09 ^g^	62.19 ± 0.13 ^l^	8.30 ± 0.17 ^gh^	14.16 ± 0.11 ^h^	5.81 ± 0.10 ^f^	27.40 ± 0.16 ^j^
	Bud block	8.01 ± 0.07 ^h^	58.29 ± 0.12 ^n^	7.27 ± 0.16 ^i^	13.24 ± 0.11 ^k^	4.90 ± 0.09 ^i^	27.89 ± 0.18 ^i^
Youjin	Top bud	12.08 ± 0.09 ^a^	78.12 ± 0.06 ^a^	9.86 ± 0.14 ^de^	15.70 ± 0.08 ^d^	7.38 ± 0.06 ^a^	32.56 ± 0.14 ^a^
	Middle bud	11.87 ± 0.10 ^a^	76.46 ± 0.18 ^c^	9.54 ± 0.23 ^ef^	14.96 ± 0.06 ^f^	7.02 ± 0.11 ^b^	31.28 ± 0.17 ^b^
	Tail bud	11.20 ± 0.12 ^b^	73.57 ± 0.19 ^e^	8.12 ± 0.13 ^h^	13.87 ± 0.10 ^i^	6.71 ± 0.15 ^c^	28.90 ± 0.12 ^ef^
	Bud block	10.57 ± 0.12 ^d^	70.18 ± 0.10 ^g^	7.14 ± 0.17 ^i^	12.74 ± 0.10 ^l^	5.84 ± 0.15 ^f^	29.02 ± 0.12 ^de^
Zhongshu 4	Top bud	10.56 ± 0.17 ^d^	70.18 ± 0.10 ^g^	10.84 ± 0.16 ^a^	16.80 ± 0.12 ^a^	6.27 ± 0.15 ^d^	29.04 ± 0.11 ^de^
	Middle bud	10.03 ± 0.11 ^f^	68.57 ± 0.22 ^h^	10.41 ± 0.23 ^bc^	16.06 ± 0.05 ^c^	6.17 ± 0.08 ^de^	28.24 ± 0.15 ^h^
	Tail bud	9.07 ± 0.10 ^g^	65.29 ± 0.12 ^k^	9.56 ± 0.27 ^ef^	14.67 ± 0.08 ^g^	5.74 ± 0.23 ^fg^	27.18 ± 0.13 ^j^
	Bud block	8.16 ± 0.09 ^h^	60.44 ± 0.14 ^m^	8.27 ± 0.15 ^gh^	13.49 ± 0.12 ^j^	5.32 ± 0.11 ^h^	27.45 ± 0.17 ^j^
Feiwurita	Top bud	11.35 ± 0.07 ^b^	77.20 ± 0.13 ^b^	10.68 ± 0.10 ^ab^	16.29 ± 0.10 ^b^	6.39 ± 0.11 ^d^	30.98 ± 0.18 ^b^
	Middle bud	10.94 ± 0.08 ^c^	75.03 ± 0.09 ^d^	10.15 ± 0.12 ^cd^	15.40 ± 0.09 ^e^	5.27 ± 1.35 ^h^	29.45 ± 0.19 ^c^
	Tail bud	10.18 ± 0.09 ^ef^	72.14 ± 0.12 ^f^	9.36 ± 0.13 ^f^	14.02 ± 0.07 ^hi^	5.94 ± 0.10 ^ef^	28.43 ± 0.17 ^gh^
	Bud block	9.25 ± 0.14 ^g^	68.04 ± 0.11 ^i^	8.49 ± 0.09 ^g^	12.62 ± 0.12 ^l^	5.47 ± 0.18 ^gh^	28.63 ± 0.13 ^fg^

Means followed by a different letter within the column are significantly different at (*p* < 0.05) probability level according to the analysis of variance (ANOVA).

**Table 4 plants-12-03545-t004:** Effects of different lengths of potato buds on plant growth.

Treatment		Plant Height (cm)		Stem Diameter (mm)		Number of Compound Leaves (Piece)	
		Seedling Stage	Expansion Period	Seedling Stage	Expansion Period	Seedling Stage	Expansion Period
Fujin	Bud length 10 cm	9.15 ± 0.15 ^g^	61.17 ± 2.3 ^hi^	9.25 ± 0.11 ^h^	14.01 ± 0.4 ^d^	5.72 ± 0.16 ^ef^	28.75 ± 0.13 ^gh^
	Bud length 15 cm	10.27 ± 0.09 ^f^	67.89 ± 3.2 ^fg^	10.14 ± 0.14 ^f^	16.74 ± 0.8 ^abc^	6.34 ± 0.18 ^d^	29.36 ± 0.19 ^f^
	Bud length 20 cm	11.09 ± 0.09 ^d^	75.15 ± 2.8 ^cde^	11.04 ± 0.16 ^cd^	17.15 ± 0.8 ^ab^	7.21 ± 0.16 ^bc^	30.45 ± 0.16 ^e^
	Bud block	8.01 ± 0.07 ^h^	58.29 ± 1.8 ^i^	7.27 ± 0.10 ^k^	13.24 ± 0.5 ^de^	4.90 ± 0.12 ^h^	27.89 ± 0.18 ^i^
Youjin	Bud length 10 cm	11.27 ± 0.15 ^cd^	73.24 ± 2.5 ^cdef^	8.77 ± 0.09 ^i^	13.62 ± 0.5 ^de^	6.54 ± 0.14 ^d^	31.14 ± 0.05 ^d^
	Bud length 15 cm	12.08 ± 0.09 ^b^	78.12 ± 2.0 ^bc^	9.86 ± 0.12 ^g^	15.70 ± 0.4 ^c^	7.38 ± 0.07 ^b^	32.56 ± 0.14 ^b^
	Bud length 20 cm	13.14 ± 0.16 ^a^	84.27 ± 2.4 ^a^	11.25 ± 0.15 ^c^	16.23 ± 0.7 ^bc^	8.02 ± 0.10 ^a^	33.27 ± 0.16 ^a^
	Bud block	10.57 ± 0.07 ^e^	70.18 ± 2.4 ^efg^	7.14 ± 0.14 ^k^	12.74 ± 0.4 ^e^	5.84 ± 0.12 ^e^	29.02 ± 0.12 ^fg^
Zhongshu 4	Bud length 10 cm	9.07 ± 0.09 ^g^	65.34 ± 2.2 ^gh^	9.60 ± 0.15 ^g^	14.12 ± 0.3 ^d^	5.88 ± 0.14 ^e^	28.45 ± 0.08 ^h^
	Bud length 15 cm	10.56 ± 0.16 ^e^	70.18 ± 2.1 ^efg^	10.84 ± 0.09 ^de^	16.80 ± 0.4 ^abc^	6.27 ± 0.13 ^d^	29.04 ± 0.11 ^fg^
	Bud length 20 cm	11.47 ± 0.17 ^c^	76.80 ± 2.6 ^bcd^	12.06 ± 0.09 ^a^	17.56 ± 0.4 ^a^	7.03 ± 0.15 ^c^	31.69 ± 0.13 ^c^
	Bud block	8.16 ± 0.11 ^h^	60.24 ± 3.0 ^hi^	8.27 ± 0.15 ^j^	13.49 ± 0.4 ^de^	5.32 ± 0.11 ^g^	27.45 ± 0.17 ^j^
Feiwurita	Bud length 10 cm	10.04 ± 0.12 ^f^	72.15 ± 2.1 ^def^	9.09 ± 0.11 ^jh^	13.80 ± 0.2 ^de^	5.98 ± 0.07 ^e^	29.75 ± 0.20 ^f^
	Bud length 15 cm	11.35 ± 0.06 ^cd^	77.2 ± 2.0 ^bcd^	10.68 ± 0.12 ^e^	16.29 ± 0.4 ^bc^	6.39 ± 0.09 ^d^	30.98 ± 0.18 ^d^
	Bud length 20 cm	12.1 ± 0.16 ^b^	81.56 ± 2.6 ^ab^	11.72 ± 0.14 ^b^	17.43 ± 0.4 ^ab^	7.08 ± 0.10 ^c^	31.56 ± 0.22 ^c^
	Bud block	9.25 ± 0.14 ^g^	68.04 ± 1.5 ^fg^	8.49 ± 0.10 ^j^	12.62 ± 0.3 ^e^	5.47 ± 0.14 ^fg^	28.63 ± 0.13 ^h^

Means followed by a different letter within the column are significantly different at (*p* < 0.05) probability level according to the analysis of variance (ANOVA).

**Table 5 plants-12-03545-t005:** Effects of different positions of sweet potato on plant photosynthetic parameters.

Treatments		Chlorophyll Content (mg/g)		Photosynthetic Rate (Pn) (μmolCO_2_·m^−2^·s^−1^)	
		Seedling Stage	Expansion Period	Seedling Stage	Expansion Period
Fujin	Top bud	2.25 ± 0.15 ^bc^	3.92 ± 0.07 ^b^	12.98 ± 0.06 ^e^	19.21 ± 0.12 ^e^
	Middle bud	2.17 ± 0.12 ^bcd^	3.80 ± 0.08 ^bc^	12.60 ± 0.11 ^ef^	18.82 ± 0.09 ^g^
	Tail bud	1.92 ± 0.07 ^e^	3.42 ± 0.10 ^de^	11.39 ± 0.06 ^h^	17.90 ± 0.08 ^j^
	Bud block	1.40 ± 0.10 ^fg^	3.06 ± 0.07 ^f^	10.63 ± 0.10 ^j^	16.31 ± 0.07 ^l^
Youjin	Top bud	2.17 ± 0.11 ^bcd^	3.80 ± 0.12 ^bc^	12.04 ± 0.08 ^g^	19.01 ± 0.07 ^f^
	Middle bud	1.98 ± 0.06 ^de^	3.60 ± 0.10 ^cd^	12.71 ± 0.09 ^f^	18.63 ± 0.07 ^h^
	Tail bud	1.60 ± 0.09 ^f^	3.34 ± 0.06 ^e^	11.34 ± 0.10 ^h^	17.44 ± 0.09 ^k^
	Bud block	1.32 ± 0.05 ^g^	2.80 ± 0.11 ^g^	10.14 ± 0.09 ^k^	15.80 ± 0.07 ^m^
Zhongshu 4	Top bud	2.84 ± 0.09 ^a^	4.52 ± 0.07 ^a^	13.87 ± 0.16 ^c^	19.71 ± 0.07 ^c^
	Middle bud	2.66 ± 0.05 ^a^	4.37 ± 0.12 ^a^	13.28 ± 0.18 ^d^	19.40 ± 0.07 ^d^
	Tail bud	2.13 ± 0.12 ^cde^	3.97 ± 0.06 ^b^	12.23 ± 0.14 ^g^	18.52 ± 0.12 ^hi^
	Bud block	1.59 ± 0.10 ^f^	3.31 ± 0.07 ^e^	10.98 ± 0.11 ^i^	17.33 ± 0.09 ^k^
Feiwurita	Top buds	2.39 ± 0.15 ^b^	3.94 ± 0.09 ^b^	14.62 ± 0.16 ^a^	21.34 ± 0.08 ^a^
	Middle bud	2.17 ± 0.12 ^bcd^	3.80 ± 0.08 ^bc^	14.21 ± 0.14 ^b^	20.81 ± 0.09 ^b^
	Tail bud	1.92 ± 0.08 ^e^	3.50 ± 0.12 ^de^	13.20 ± 0.13 ^de^	19.90 ± 0.15 ^c^
	Bud block	1.47 ± 0.11 ^fg^	2.98 ± 0.05 ^fg^	11.19 ± 0.13 ^hi^	18.41 ± 0.09 ^i^

Means followed by a different letter within the column are significantly different at (*p* < 0.05) probability level according to the analysis of variance (ANOVA).

**Table 6 plants-12-03545-t006:** Effects of different potato bud lengths on plant photosynthetic parameters.

Treatment		Chlorophyll Content (mg/g)		Photosynthetic Rate (Pn) (μmolCO_2_·m^−2^·s^−1^)	
		Seedling Stage	Expansion Period	Seedling Stage	Expansion Period
Fujin	Bud length 10 cm	1.92 ± 0.08 ^hi^	3.37 ± 0.12 ^f^	11.30 ± 0.06 ^g^	18.11 ± 0.22 ^e^
	Bud length 15 cm	2.25 ± 0.11 ^efg^	3.92 ± 0.17 ^de^	12.98 ± 0.04 ^d^	19.20 ± 0.15 ^cd^
	Bud length 20 cm	2.63 ± 0.08 ^bcd^	4.21 ± 0.06 ^c^	13.78 ± 0.09 ^e^	19.60 ± 0.23 ^c^
	Bud block	1.40 ± 0.14 ^k^	3.06 ± 0.08 ^h^	10.63 ± 0.06 ^i^	16.31 ± 0.13 ^g^
Youjin	Bud length 10 cm	1.84 ± 0.14 ^ij^	3.14 ± 0.11 ^gh^	11.28 ± 0.13 ^gh^	17.32 ± 0.18 ^f^
	Bud length 15 cm	2.17 ± 0.11 ^fgh^	3.80 ± 0.08 ^e^	12.04 ± 0.04 ^f^	19.01 ± 0.18 ^d^
	Bud length 20 cm	2.50 ± 0.13 ^cde^	4.05 ± 0.07 ^cd^	12.97 ± 0.05 ^e^	19.44 ± 0.09 ^cd^
	Bud block	1.32 ± 0.15 ^k^	2.80 ± 0.11 ^i^	10.14 ± 0.06 ^j^	15.82 ± 0.13 ^h^
Zhongshu 4	Bud length 10 cm	2.32 ± 0.15 ^ef^	3.95 ± 0.06 ^de^	12.05 ± 0.08 ^f^	18.90 ± 0.11 ^d^
	Bud length 15 cm	2.84 ± 0.15 ^ab^	4.52 ± 0.04 ^b^	13.87 ± 0.08 ^d^	19.71 ± 0.59 ^c^
	Bud length 20 cm	3.07 ± 0.09 ^a^	4.74 ± 0.14 ^a^	14.92 ± 0.04 ^b^	20.33 ± 0.19 ^b^
	Bud block	1.59 ± 0.19 ^jk^	3.31 ± 0.15 ^fg^	10.98 ± 0.12 ^h^	17.34 ± 0.20 ^f^
Feiwurita	Bud length 10 cm	2.01 ± 0.07 ^ghi^	3.40 ± 0.10 ^f^	13.15 ± 0.09 ^e^	18.93 ± 0.21 ^d^
	Bud length 15 cm	2.39 ± 0.15 ^def^	3.94 ± 0.06 ^de^	14.62 ± 0.11 ^c^	21.34 ± 0.18 ^a^
	Bud length 20 cm	2.78 ± 0.17 ^abc^	4.12 ± 0.04 ^cd^	15.84 ± 0.05 ^a^	21.70 ± 0.23 ^a^
	Bud block	1.47 ± 0.10 ^k^	2.98 ± 0.06 ^hi^	11.19 ± 0.08 ^gh^	18.44 ± 0.13 ^e^

Means followed by a different letter within the column are significantly different at (*p* < 0.05) probability level according to the analysis of variance (ANOVA).

**Table 7 plants-12-03545-t007:** Effects of different bud positions on potato growing time.

Treatment		Sowing Time (Month/Day)	Harvest Time (Month/Day)	Growing Days in Shed (Day)	Relative Growing Days
Fujin	Top bud	4/20	6/17	57	−15
	Middle bud	4/20	6/19	59	−13
	Tail bud	4/20	7/5	75	+3
	Bud block	4/20	7/2	72	0
Youjin	Top bud	4/20	6/20	60	−14
	Middle bud	4/20	6/24	64	−10
	Tail bud	4/20	7/15	78	+4
	Bud blocks	4/20	7/11	74	0
Zhongshu 4	Top bud	4/20	6/15	55	−14
	Middle bud	4/20	6/18	58	−11
	Tail bud	4/20	7/8	71	+2
	Bud blocks	4/20	7/6	69	0
Feiwurita	Top bud	4/20	6/22	62	−11
	Middle bud	4/20	7/1	64	−9
	Tail bud	4/20	7/12	75	+2
	Bud block	4/20	7/10	73	0

**Table 8 plants-12-03545-t008:** The influence of the different parts of the potato buds used on yield by direct cultivation.

Treatment		Number of Tubers per Plant (PCs.)	Average Weight per Potato (g)	Average Plot Yield (kg)	Yield (kg/ha)	Relative Yield %
Fujin	Top bud	5.0 ^a^	135.28 ^def^	23.81 ^f^	3.97 × 10^4 f^	122.66
	Middle bud	5.0 ^a^	132.78 ^f^	23.37 ^g^	3.90 × 10^4 h^	120.40
	Tail bud	5.0 ^a^	117.44 ^h^	21.14 ^j^	3.53 × 10^4 k^	109.20
	Bud blocks	4.0 ^b^	121.31 ^h^	19.4 ^l^	3.24 × 10^4 m^	100.00
Youjin	Top bud	5.0 ^a^	143.22 ^c^	25.7 ^b^	4.30 × 10^4 b^	129.23
	Middle bud	5.0 ^a^	142.56 ^c^	25.09 ^d^	4.18 × 10^4 d^	125.80
	Tail bud	5.0 ^a^	121.39 ^h^	21.85 ^i^	3.64 × 10^4 j^	109.57
	Bud blocks	4.0 ^b^	127.88 ^g^	19.95 ^k^	3.33 × 10^4 l^	100.00
Zhongshu 4	Top bud	5.0 ^a^	150.63 ^b^	26.51 ^a^	4.42 × 10^4 a^	138.05
	Middle bud	4.0 a	158.38 ^a^	25.34 ^c^	4.23 × 10^4 c^	131.94
	Tail bud	4.0 ^b^	138.31 ^d^	22.13 ^h^	3.69 × 10^4 i^	115.24
	Bud blocks	5.0 ^a^	126.97 ^g^	19.21 ^l^	3.20 × 10^4 n^	100.00
Feiwurita	Top bud	5.0 ^a^	137.28 ^de^	24.71 ^e^	4.12 × 10^4 e^	136.11
	Middle bud	4.0 ^b^	133.86 ^ef^	23.56 ^g^	3.93 × 10^4 g^	129.81
	Tail bud	4.0 ^b^	136.56 ^def^	21.85 ^i^	3.64 × 10^4 j^	120.39
	Bud block	5.0 ^a^	119.41 ^h^	18.15 ^m^	3.03 × 10^4 o^	100.00

Means followed by a different letter within the column are significantly different at (*p* < 0.05) probability level according to the analysis of variance (ANOVA).

**Table 9 plants-12-03545-t009:** Effects of different bud lengths on potato growing time.

Treatment		Sowing Time (Month/Day)	Harvest Time (Month/Day)	Growing Days in Shed (Day)	Relative Growing Days
Fujin	Bud length 10 cm	4/20	6/21	61	−11
	Bud length 15 cm	4/20	6/17	57	−15
	Bud length 20 cm	4/20	6/16	56	−16
	Bud block	4/20	7/2	72	0
Youjin	Bud length 10 cm	4/20	6/24	64	−10
	Bud length 15 cm	4/20	6/20	60	−14
	Bud length 20 cm	4/20	6/18	58	−16
	Bud block	4/20	7/4	74	0
Zhongshu 4	Bud length 10 cm	4/20	6/18	58	−11
	Bud length 15 cm	4/20	6/15	55	−14
	Bud length 20 cm	4/20	6/14	54	−15
	Bud block	4/20	6/29	69	0
Feiwurita	Bud length 10 cm	4/20	6/28	68	−5
	Bud length 15 cm	4/20	6/22	62	−11
	Bud length 20 cm	4/20	6/20	60	−13
	Bud block	4/20	7/3	73	0

**Table 10 plants-12-03545-t010:** Effects of different bud lengths on potato yield.

Treatment		Number of Tubers per Plant (PCs.)	Average Weight per Potato (g)	Average Plot Yield (kg)	Yield (kg/ha)	Relative Yield %
Fujin	Bud length 10 cm	4.0 ^b^	125.25 ^fg^	20.04 ^i^	3.66 × 10^4 h^	113.00
	Bud length 15 cm	4.0 ^b^	135.28 ^e^	23.81 ^fg^	3.97 × 10^4 fg^	122.66
	Bud length 20 cm	5.0 ^a^	135.28 ^e^	24.26 ^e^	4.08 × 10^4 ef^	126.00
	Bud block	4.0 ^b^	121.31 ^gh^	19.41 ^j^	3.24 × 10^4 j^	100.00
Youjin	Bud length 10 cm	4.0 ^b^	148.38 ^a^	23.74 ^g^	3.96 × 10^4 g^	119.01
	Bud length 15 cm	5.0 ^a^	143.22 ^bc^	25.78 ^c^	4.30 × 10^4 bc^	129.23
	Bud length 20 cm	5.0 ^a^	147.33 ^ab^	26.52 ^b^	4.42 × 10^4 b^	132.95
	Bud block	4.0 ^b^	127.88 ^f^	19.95 ^i^	3.33 × 10^4 i^	100.00
Zhongshu 4	Bud length 10 cm	4.0 ^b^	151.06 ^a^	24.17 ^ef^	4.03 × 10^4 f^	125.85
	Bud length 15 cm	5.0 ^a^	150.63 ^a^	26.51 ^b^	4.42 × 10^4 ab^	138.05
	Bud length 20 cm	5.0 ^a^	151.28 ^a^	27.23 ^a^	4.54 × 10^4 a^	141.78
	Bud block	4.0 ^b^	126.97 ^f^	19.21 ^j^	3.20 × 10^4 j^	100.00
Feiwurita	Bud length 10 cm	4.0 ^b^	136.63 ^e^	21.86 ^h^	3.65 × 10^4 h^	120.44
	Bud length 15 cm	5.0 ^a^	137.28 ^de^	24.71 ^d^	4.12 × 10^4 de^	136.11
	Bud length 20 cm	5.0 ^a^	141.28 ^cd^	25.43 ^c^	4.24 × 10^4 d^	140.08
	Bud block	4.0 ^b^	119.41 ^h^	18.15 ^k^	3.03 × 10^4 k^	100.00

Means followed by a different letter within the column are significantly different at (*p* < 0.05) probability level according to the analysis of variance (ANOVA).

**Table 11 plants-12-03545-t011:** The influence of the different parts of the potato buds on quality by direct cultivation.

Treatment		Starch (%)	Reducing Sugar (%)	Crude Protein (%)	Vc Content (mg/100 g)	Commercial Potato Growth Rate %
Fujin	Top bud	14.36 ^a^	0.088 ^c^	1.708 ^a^	0.274 ^c^	87.8
	Middle bud	14.33 ^a^	0.086 ^c^	1.702 ^a^	0.273 ^c^	86.3
	Tail bud	14.22 ^a^	0.085 ^c^	1.695 ^a^	0.271 ^c^	84.7
	Bud blocks	14.16 ^a^	0.083 ^c^	1.687 ^a^	0.269 ^c^	82.5
Youjin	Top bud	13.55 ^b^	0.063 ^d^	1.503 c	0.275 ^c^	92.0
	Middle bud	13.50 ^b^	0.062 ^d^	1.496 ^c^	0.274 ^c^	81.7
	Tail bud	13.46 ^b^	0.062 ^d^	1.492 ^c^	0.272 ^c^	89.9
	Bud blocks	13.43 ^b^	0.061 ^d^	1.487 ^c^	0.271 ^c^	88.7
Zhongshu 4	Top bud	12.54 ^c^	0.441 ^b^	1.604 ^b^	26.713 ^b^	87.7
	Middle bud	12.50 ^c^	0.438 ^b^	1.599 ^b^	26.654 ^b^	86.0
	Tail bud	12.46 ^c^	0.436 ^b^	1.594 ^b^	26.595 ^b^	83.2
	Bud blocks	12.41 ^cd^	0.429 ^b^	1.589 ^b^	26.492 ^b^	80.8
Feiwurita	Top bud	12.42 ^cd^	1.021 ^a^	1.019 ^d^	113.101 ^a^	89.1
	Middle bud	12.40 ^cd^	1.018 ^a^	1.016 ^d^	112.915 ^a^	88.4
	Tail bud	12.38 ^cd^	1.013 ^a^	1.014 ^d^	112.762 ^a^	87.0
	Bud block	12.35 ^cd^	1.009 ^a^	1.011 ^d^	112.623 ^a^	85.0

Means followed by a different letter within the column are significantly different at (*p* < 0.05) probability level according to the analysis of variance (ANOVA).

**Table 12 plants-12-03545-t012:** The influence of the different lengths of the potato buds on quality by direct cultivation.

Treatment		Starch (%)	Reducing Sugar (%)	Crude Protein (%)	Vc Content (mg/100 g)	Commercial Potato Growth Rate %
Fujin	Bud length 10 cm	14.22 ^a^	0.085 ^c^	1.696 ^a^	0.273 ^c^	84.1
	Bud length 15 cm	14.36 ^a^	0.088 ^c^	1.708 ^a^	0.274 ^c^	87.8
	Bud length 20 cm	14.42 ^a^	0.089 ^c^	1.714 ^a^	0.280 ^c^	86.6
	Bud block	14.16 ^a^	0.083 ^c^	1.687 ^a^	0.269 ^c^	82.5
Youjin	Bud length 10 cm	13.45 ^b^	0.062 ^d^	1.495 ^c^	0.272 ^c^	89.9
	Bud length 15 cm	13.55 ^b^	0.063 ^d^	1.503 ^c^	0.275 ^c^	92.0
	Bud length 20 cm	13.60 ^b^	0.064 ^d^	1.510 ^c^	0.281 ^c^	91.2
	Bud block	13.43 ^b^	0.061 ^d^	1.487 ^c^	0.271 ^c^	88.7
Zhongshu 4	Bud length 10 cm	12.49 ^c^	0.436 ^b^	1.598 ^b^	26.601 ^b^	83.2
	Bud length 15 cm	12.54 ^c^	0.441 ^b^	1.604 ^b^	26.710 ^b^	87.7
	Bud length 20 cm	12.58 ^c^	0.443 ^b^	1.610 ^b^	27.021 ^b^	86.8
	Bud block	12.41 ^cd^	0.429 ^b^	1.589 ^b^	26.492 ^b^	80.8
Feiwurita	Bud length 10 cm	12.36 ^d^	1.010 ^a^	1.014 ^d^	112.873 ^a^	87.0
	Bud length 15 cm	12.42 ^cd^	1.021 ^a^	1.019 ^d^	113.101 ^a^	89.1
	Bud length 20 cm	12.47 ^cd^	1.023 ^a^	1.021 ^d^	113.541 ^a^	88.3
	Bud block	12.35 ^cd^	1.009 ^a^	1.011 ^d^	112.623 ^a^	85.0

Means followed by a different letter within the column are significantly different at (*p* < 0.05) probability level according to the analysis of variance (ANOVA).

## Data Availability

Not applicable.

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
