# Peer review of "Study on Potato Bud Cultivation Techniques in a Greenhouse in Spring"

_plants, 2023, doi:10.3390/plants12203545_

Round 1

Reviewer 1 Report

My general opinion about the manuscript is positive, and for this reason I recommend it for publication in the Plants.

Author Response

Dear reviewer:

First of all, thank you very much for taking time out of your busy schedule to review our manuscript and give positive comments. As for the problem of English quality you mentioned, we have re-polished the whole manuscript. Finally, we submitted the revised manuscript, including one with modified traces and one traceless final version, for your review.

We would like to express our great appreciation to you again for comments on our paper. 

Best regards.

Yours sincerely,

Chong Du

Reviewer 2 Report

Dear Authors,

The manuscript described the results of the greenhouse and the laboratory experiment on the Study on potato bud cultivation techniques in a greenhouse in spring. The subject is interesting, however an article’s form, exactly: aim, materials and methods, results, conclusions should be changed in their whole.

Manuscript cannot be accepted for publishing in the Plants journal in this form. The results are from a single experiment. This is unacceptable. Experiments should be conducted for a minimum of 2 years and in repetitions.

There is no results values in the abstract.

The aim of the research was incorrectly formulated. It's not readable.

Lack Supplementary Figures: S1, S1A, and S1B.

What do the letters next to the results mean in tables 1-6, 8, 10-12.

There is no information about what statistical methods were used to properly discuss the results.

Are the results of chlorophyll, starch, sugars and vitamin C content in fresh or dry matter?

What methods were used to study the chemical and physical parameters of the soil?

The method for determining the chlorophyll and dry matter content in potatoes has not been described?

Four potato cultivars were used in the experiment. Do all cultivars belong to the same earliness group?

Author Response

Dear reviewer:

First of all, thank you very much for taking time out of your busy schedule to review our manuscript and give valuable suggestions. As for the problem of English quality you mentioned, we have re-polished the whole manuscript. According to the questions you raised, we conducted one-to-one corrections and replies. Finally, we submitted the revised manuscript, including one with modified traces and one traceless final version, for your review.

According to the requirements, we have made a lot of modifications to make the purpose of the study more clear, the description of the materials, methods more comprehensive and easy to read, and the results, conclusions more detailed and accurate.

  1. We are very sorry for the trouble caused to you by our negligence and wrong description. This study, conducted from March to July in 2022 and 2023, respectively, meets the basic requirements of the experiment, and we have added this part to the section of ‘materials and methods’. (line 404-405)
  2. Thanks to your suggestion, we have revised the description of abstract and added important values that reflect the research results. (line 18-27)
  3. We are very sorry for our incorrect writing. We have supplemented and retouched up the purpose of the study. (line 70-81)
  4. Considering the reviewer's suggestion, we have supplemented Figures S1 and S2 in the folder where we submitted the revised manuscript.
  5. These letters represent the magnitude of the significant difference, and at the reviewer's request, we have added a note to the end of the table referring to the representation of the significant difference. (line 104-105, line 115-116, line 138-139, line 162-163, line 186-187, line 210-211, line 281-282, line 325-326, line 349-350, line 352-353)
  6. Thanks to the reviewer's valuable comments, we added a subsection to the ‘methods’ section, describing the statistical methods and the software used, hoping to improve the readability of the manuscript. (line 455-459)
  7. The quality indicators in this study, including chlorophyll, starch, sugars and vitamin C, were all determined in fresh matter. Therefore, according to the reviewer's comments, we have modified this part of the description in the sections of ‘results’ and ‘methods’, respectively. (line 328 and line 448-449)
  8. According to the reviewer's requirements, we added the references ([31]-[34]) of methods for determining the physical and chemical properties of soil in the part of ‘methods’. (line 406-408)
  9. According to the reviewer's opinion, regarding the method of quality indicator determination, we have introduced the references ([36]-[41]) used in the ‘method’ section. (line 439-454)
  10. It is true that all 4 tested cultivars belong to the same earliness group, according to reviewer' suggestion, specific information about the 4 potato varieties is described and added to the ‘materials’. (line 399-404)

We would like to express our great appreciation to you again for comments on our paper. We hope our modification can meet the publication requirements.

Best regards.

Yours sincerely,

Chong Du

Reviewer 3 Report

The authors have comprehensively compared the growth, yield and quality related traits of four potato varieties with bud direct seeding under three parts and three bud lengths. The results suggest that direct seeding with longer potato sprouts (20 cm) can achieve higher emergence rate, leaf number, photosynthetic efficiency, dry matter distribution of tubers, early maturity, and yield, which is very meaningful for farmers and agronomists to a certain extent. However, some advice as below need to be further revised when the manuscript gets published in this journal.

1. Introduction: Please divide the first paragraph into paragraphs.

2. Lines 184-185 and lines 197-198: Please supplement the description of T1, T2, T3, T4, T5, T6 and CK in Figure 1 and Figure 2.

3. Line 171: It is better to describe this context about “The dry matter allocation rate of potato plants directly cultivated from potato buds taken from different parts during the potato block expansion period” in the first part, and later describe “The distribution rate of dry matter in potato plants during the expansion period of potato block enlargement with different lengths of potato top-buds directly planted”. Figure 1 and Figure 2 also need to be change position each other.

4. Lines 190-192: This sentence has no comparison among treatments, please delete “compared to the control treatment”.

5. Please supplement some latest and global references in “Introduction” and “Discussion” parts.

Author Response

Dear reviewer:

First of all, thank you very much for taking time out of your busy schedule to review our manuscript and give valuable suggestions. As for the problem of English quality you mentioned, we have re-polished the whole manuscript. According to the questions you raised, we conducted one-to-one corrections and replies. Finally, we submitted the revised manuscript, including one with modified traces and one traceless final version, for your review.

  1. Considering the reviewer's suggestion, we have divided the first paragraph into paragraphs so as to improve the readability of this paper. (line 40 and line 56)
  2. We are very sorry for the wrong description in our figures. According to the reviewer's suggestion, we have replaced the description processed in the Figure 1 and 2
  3. Thanks for the valuable suggestion, we have adjusted the contents of part 2.4 of the ‘result’ The results of ‘dry matter distribution in different parts’ are put in the front, and the results of ‘dry matter distribution in different lengths’ are put in the back, and the corresponding picture order is adjusted. (line 213-222 and line 226-236)
  4. We are very sorry for our incorrect writing, we have deleted ‘compared to the control treatment’. (line 218)
  5. Thank you for your valuable advice, because in the introduction part, our research is based on the previous achievements of Chinese scholars, so we left some classical literatures without modification. In the discussion part, we have updated the literature comprehensively to make the research value more up-to-date and improve the quality of the paper.(line 354-396, line 485-568)

We would like to express our great appreciation to you again for comments on our paper. We hope our modification can meet the publication requirements.

Best regards.

Yours sincerely,

Chong Du

Reviewer 4 Report

The following are my comments, below:

Abstract

-Very long sentence: Please revise” By comprehensively measuring the emergence, chlorophyll content and net photosynthetic rate, dry matter distribution during the bulking period of blocks, effect of growth, early maturity and yield, and quality of potato with bud direct seeding under both treatments, in summary, it was found that potato block top bud direct seeding cultivation is significantly superior to other parts and is more suitable for potato sprout planting technology.”

-The impact of the research is not clear: At the end of the abstract, please show what is the message for the potato growers and how they can benefit from your research.

Materials and Methods:

I did not see any experimental design and statistical analysis (number of replicates, software used for statistical analysis, design of the experiment, and so on). I would suggest that the authors create a sub-section called “Experimental Design and Statistical Analysis” and add the information under this section.

References

The following references are not consistent in their title font type (capital letter and small letter. Please be consistent with one type throughout  the manuscript.

16. Wang, C.; Du, C.; Yang, Z.; Wang, H.; Shang, L.; Liu, L.; Yang, Z.; Song, S. and Amanullah, S. Study on the cultivation of seedlings 405 using buds of potato (Solanum tuberosum L.). PeerJ. 2022, 10. 406

17. He, T. Technology of transplanting and propagation of Potato by breaking seedlings for many times. China Potato. 1997, 1, 25– 407 26 (in Chinese). 408

18. Zhou, H.; Wu, X.; Li, W. Study on the effect of different seedling raising methods on cucumber yield in autumn. Bulletin of 409 Agricultural Science and Technology. 2014, 2, 109–110 (in Chinese). 410

19. Long, G.; Zhang, S.; Cao, X.; et al. Studies on the Effect of Different Bud Position Cutting Patato Seed on Related Characters. 411 Seed. 2009, 28(10), 97-99 (in Chinese).

27. Zhou, H.; Wu, X.; Li, W. Study on the Effect of Different Seedling Raising Methods on Cucumber Yield in Autumn Open Field. 425 Bulletin of Agricultural Science and Technology. 2014, 2, 109-110 (in Chinese). 426 28.

28. Zhang, Y.; Tian, F. Study on physiological characteristics of high yield and good quality of potato. China Agricultural Science and 427 Technology Press. 2012 (in Chinese).

Very long sentence: Please revise” By comprehensively measuring the emergence, chlorophyll content and net photosynthetic rate, dry matter distribution during the bulking period of blocks, effect of growth, early maturity and yield, and quality of potato with bud direct seeding under both treatments, in summary, it was found that potato block top bud direct seeding cultivation is significantly superior to other parts and is more suitable for potato sprout planting technology.”

Author Response

Dear reviewer:

First of all, thank you very much for taking time out of your busy schedule to review our manuscript and give valuable suggestions. As for the problem of English quality you mentioned, we have re-polished the whole manuscript. According to the questions you raised, we conducted one-to-one corrections and replies. Finally, we submitted the modified version, including one with modified traces and one traceless final version, for your review.

  1. Thank you for your suggestion. We have re-written this part, we added specific numerical values to display the important results, simplified the complex sentences (in traceless final version) and added a summary description at the end of the abstract to clarify the impact of this study. (line 7-27)
  2. Thanks to the reviewer's valuable comments, we added a subsection to the ‘methods’section, describing the statistical methods and the software used, hoping to improve the readability of the manuscript. (line 455-459)
  3. We are very sorry for our miswriting of references. According to your comments, we have checked the references and corrected the incorrect description. (line 485-568)

We would like to express our great appreciation to you again for comments on our paper. We hope our modification can meet the publication requirements.

Best regards.

Yours sincerely,

Chong Du

Round 2

Reviewer 2 Report

I accept the article.

Reviewer 4 Report

Dear Editor,

The authors responded to my  comments appropriately, and their response improved the quality of the manuscript. Therefore, I am recommending the manuscript for acceptance and publication. Thank you for the opportunity.